# FATIGUE-AWARE LEARNING TO DEFER VIA CONSTRAINED OPTIMISATION

## ABSTRACT

Learning to defer (L2D) enables human-AI cooperation by determining when AI systems should make autonomous predictions versus deferring to human experts. However, existing L2D methods assume constant human performance across both short and long time horizons, contradicting established cognitive psychology research on fatigue-induced performance degradation. We present Fatigue-Aware Learning to Defer via Constrained Optimisation (FALCON), explicitly modelling workload-varying human performance through psychologically grounded fatigue curves. FALCON formulates L2D as a Constrained Markov Decision Process (CMDP), where system states incorporate both task-specific characteristics and cumulative human workload. In particular, we maximise classification accuracy under human-AI cooperation budget constraints, using PPO-Lagrangian optimisation. We also introduce the Fatigue-Aware L2D (FA-L2D) benchmark with controllable fatigue-induced performance degradation across varying time horizons, enabling scenarios that range from near-constant to highly variable human performance and replacing prior benchmarks that assumed stability over time. Extensive experiments on our benchmarks demonstrate that FALCON consistently outperforms state-of-the-art L2D approaches at all coverage levels, particularly when considering human performance variations. Notably, FALCON enables zero-shot generalisation to unseen experts with different fatigue patterns. Furthermore, L2D methods are shown to consistently surpass both AI-only and human-only baselines whenever coverage lies strictly between 0 and 1, underscoring the effectiveness of adaptive human–AI collaboration in a setting closer to real-world scenarios.

## 1 INTRODUCTION

AI systems are increasingly deployed in safety-critical applications, but relying solely on AI can be dangerous because they may overlook subtle issues that only humans can interpret. In domains such as financial risk assessment (Green & Chen, 2019), breast cancer classification (Halling-Brown et al., 2020), and detecting deceptive AI-generated content (Ding et al., 2024), human experts provide essential judgment and contextual understanding that current AI models cannot replicate. While AI offers consistent and relatively reliable performance, it can still make catastrophic errors that humans are better positioned to detect. Conversely, humans can be highly trustworthy in complex scenarios, but their performance is unstable and influenced by factors such as expertise level and fatigue.

Learning to defer (L2D) aims to address these challenges by creating *hybrid intelligence* systems that dynamically allocate decisions between AI and human experts (Fügener et al., 2022). L2D methods learn a gating mechanism to defer decisions to humans on high-uncertainty cases to maximise accuracy, or leverage AI classifiers on high-confidence cases to minimise cost and reserve human effort (Madras et al., 2018). Current L2D approaches can be categorised by their architectural design, including one-stage and two-stage approaches. The one-stage approach (Mozannar & Sontag, 2020a) jointly models gating and classification functions using shared feature representations, while the two-stage approach (Madras et al., 2018) models these components separately. Recent research in L2D has extended from human-specific to human-adaptive setting, in which a system can quickly adapt and collaborate with a new human expert given prior knowledge of that expert capability. Specifically, L2D-Pop (Tailor et al., 2024) encodes a *context set* consisting of data annotated by a human expert in a *few-shot* setting and conditions for that representation to collaborate with that

Figure 1: Example of an L2D scenario illustrating workload-variant human performance in human–AI task allocation within a single episode. FALCON adapts deferral decisions based on both task difficulty and accumulated human fatigue. At $t = 1$, an easy task is handled by the AI while the human expert remains fresh. At $t = 2$, a challenging case is deferred to the human expert who has sufficient cognitive capacity. By $t = 3$, another hard task is still assigned to the human despite mild fatigue accumulation. At the final time step $t = T$, severe human fatigue leads to AI handling the task to prevent performance degradation.

expert. EA-L2D (Strong et al., 2025) simplifies that strategy further by using class-expertise of a human expert measured in the context set as the representation for that human.

Despite these successes, current L2D systems rely on an unrealistic assumption: they assume *human experts function as tireless, static oracles with constant performance*. This simplification is often adopted to facilitate the learning process and reduce modelling complexity, but comes at the cost of creating an unrealistic training and deployment scenario. In reality, cognitive psychology research (Casali et al., 2019) shows that human performance is dynamic, influenced by factors such as skill acquisition and, more critically, cognitive fatigue (Pimenta et al., 2014; Bose et al., 2019). As humans engage in demanding tasks for prolonged periods, their vigilance wanes, leading to an accuracy decline, which is known as the vigilance decrement phenomenon (Gyles et al., 2023). This static-performance assumption introduces practical inefficiencies in human-AI cooperation. Fatigue-induced performance degradation is well documented: individuals experience declining cognitive and physical capabilities as work sessions progress, making even simple tasks increasingly time-consuming and effortful (Cairns et al., 2008). This resource depletion is particularly pronounced during extended periods of demanding or repetitive work (Pimenta et al., 2014; Lee et al., 2013), where sustained attention requirements exceed natural cognitive capacity. In real-world scenarios involving extended work sessions, such as radiological screening, cognitive fatigue accumulates predictably (Waite et al., 2017; Reiner & Krupinski, 2012; Taylor-Phillips & Stinton, 2019), significantly increasing diagnostic error rates with potentially life-threatening consequences. Berlin (2000) reported a radiologist who made a critical misdiagnosis after interpreting 162 cases in a single day, which is more than triple the typical daily workload of 50 cases. Despite this, current L2D methods continue to apply fixed deferral thresholds throughout an entire session, ignoring temporal variations in human performance. Consequently, two equally complex tasks may be assigned to the same expert, once when fresh and later when fatigued, without accounting for the diminished cognitive resources.

Inspired by cognitive psychology research on mental fatigue (Estes, 2015; Newell & Rosenbloom, 2013), we introduce a dynamic L2D setting, illustrated in Fig. 1, that accounts for predictable variations in human performance, challenging the common assumption of static expert capability. We explicitly model workload-dependent human performance by linking expert accuracy to dynamic performance curves that capture both initial learning and subsequent fatigue-induced decline. To operationalise this, we introduce Fatigue-Aware Learning to Defer via Constrained Optimisation (FALCON), which formulates dynamic L2D as a Constrained Markov Decision Process (CMDP), where system states incorporate task-specific characteristics and cumulative human workload. This formulation enables our framework to make adaptive deferral decisions that align task allocation with the expert's current cognitive state, rather than assuming static capability under a predetermined human-AI collaboration budget. Our main contributions can be summarised as follows:

- **L2D with workload-variant human performance:** We introduce FALCON, the first framework for human–AI cooperative sequential decision-making that accounts for workload-variant

human performance by explicitly modelling its degradation over time. Additionally, FALCON incorporates a budget-constrained optimisation strategy, enabling precise control over target coverage while effectively balancing the accuracy–coverage trade-off.

- **Psychologically Grounded Simulation Environment:** We develop a human performance simulation environment grounded in psychological principles, offering a realistic testbed for evaluating L2D methods under workload-variant human performance conditions.
- **Fatigue-Aware L2D (FA-L2D) Benchmark:** We release the FA-L2D benchmark, based on Cifar100 (Wei et al., 2021), Flickr (Yang et al., 2017), MiceBone (Schmarje et al., 2022), and Chaoyang (Zhu et al., 2021), which models controllable fatigue effects across varying time horizons, enabling scenarios from near-constant to highly variable human performance and replacing prior benchmarks that assumed static human performance.

We evaluate FALCON against state-of-the-art L2D approaches (Mozannar & Sontag, 2020a; Madras et al., 2018; Tailor et al., 2024; Strong et al., 2025) on our proposed FA-L2D benchmark. Empirical results demonstrate that FALCON consistently outperforms existing methods, achieving higher accuracy for equivalent coverage levels across all evaluation settings. Importantly, under the workload-variant human performance proposed by our FA-L2D benchmark, L2D methods consistently outperform both AI-only and human-only decision-making for any non-trivial coverage level (i.e., between 0 and 1), highlighting the practical value of adaptive collaboration strategies.

## 2 PRELIMINARIES

### 2.1 LEARNING TO DEFER

For a $K$-way classification task, let $\mathcal{D} = \{(\mathbf{x}_i, \mathbf{y}_i)\}_{i=1}^N$ be the training set of size $N$, where $\mathbf{x}_i \in \mathcal{X} \subset \mathbb{R}^d$ denotes a $d$-dimensional input sample, and $\mathbf{y}_i \in \mathcal{Y} \subset \{0,1\}^K$ is the corresponding ground truth label. An *AI classifier* is denoted as $\mathsf{m} : \mathcal{X} \to \Delta^{K-1}$, where a human expert is represented by $\mathsf{h} : \mathcal{X} \to \Delta^{K-1}$. Traditional L2D methods contain the classifier $\mathsf{m}(\cdot)$ and a gating function $\mathsf{g}(\cdot)$. Given an input sample $\mathbf{x}$ and corresponding human prediction $\mathsf{h}(\mathbf{x})$ and ground truth label $\mathbf{y}$, the training objective is:

$$\ell(\mathsf{m}, \mathsf{g}) = \mathbb{E}_{\mathbf{x}, \mathbf{y}, \mathbf{h}} \left[ (1 - \mathsf{g}(\mathbf{x}))\mathbb{I}[\mathsf{h}(\mathbf{x}) \neq \mathbf{y}] + \mathsf{g}(\mathbf{x})\mathbb{I}[\mathsf{m}(\mathbf{x}) \neq \mathbf{y}] \right], \tag{1}$$

where $\mathbb{I}[\cdot]$ is the indicator function, $\mathsf{g}(\mathbf{x}) \in [0,1]$ denotes the probability deferring the decision to the human, while $1 - \mathsf{g}(\mathbf{x})$ represents the probability that the AI classifier makes the prediction. Since $\mathbb{I}[\cdot]$ is non-differentiable, some surrogate losses are proposed to generalise the cross-entropy loss (Verma & Nalisnick, 2022; Mozannar & Sontag, 2020b).

Critically, all existing L2D methods are built on the simplifying assumption that the performance of the human prediction $\mathsf{h}(\mathbf{x})$ is static over time, which is an assumption that ignores well-documented variations such as fatigue-induced degradation or learning effects (Estes, 2015; Leppink & Pérez-Fuster, 2019), and thus fails to reflect realistic deployment conditions.

### 2.2 MARKOV DECISION PROCESS

A Markov Decision Process (MDP) can be described by a 4-tuple $(\mathcal{S}, \mathcal{A}, \mathsf{p}, \mathsf{r})$, where $\mathcal{S}$ is the set of states called the *state space*, $\mathcal{A}$ is the set of actions called *action space*, $\mathsf{p} : \mathcal{S} \times \mathcal{A} \to \Delta(\mathcal{S})$ is the *transition dynamics* with $\Delta(\mathcal{S})$ being the probability simplex over $\mathcal{S}$, and $\mathsf{r} : \mathcal{S} \times \mathcal{A} \times \mathcal{S} \to \mathbb{R}$ is a *reward function*. A *policy* $\pi : \mathcal{S} \to \Delta(\mathcal{A})$ maps a state in $\mathcal{S}$ to a probability distribution over the actions in $\mathcal{A}$. An *optimal policy* $\pi^*$ is a policy that maximises the expected value of the discounted return $J_\mathsf{r}(\pi) = \mathbb{E}_{\mathbf{s}_0 \sim \mathcal{S}}[\sum_{t=0}^\infty \gamma^t \mathsf{r}(\mathbf{s}_t, \pi(\mathbf{s}_t), \mathbf{s}_{t+1})]$, where $\gamma \in [0,1]$ is a discount factor. The value function is defined as $V_\mathsf{r}^\pi(\mathbf{s}) = \mathbb{E}_{\tau \sim \pi}[\sum_t \gamma^t \mathsf{r}(\mathbf{s}_t, \mathbf{a}_t, \mathbf{s}_{t+1})|\mathbf{s}_0 = \mathbf{s}]$, the action-value function is defined as $Q_\mathsf{r}^\pi(\mathbf{s}, \mathbf{a}) = \mathbb{E}_{\tau \sim \pi}[\sum_t \gamma^t \mathsf{r}(\mathbf{s}_t, \mathbf{a}_t, \mathbf{s}_{t+1})|\mathbf{s}_0 = \mathbf{s}, \mathbf{a}_0 = \mathbf{a}]$ and the advantage function is defined as $A_\mathsf{r}^\pi(\mathbf{s}, \mathbf{a}) = Q_\mathsf{r}^\pi(\mathbf{s}, \mathbf{a}) - V_\mathsf{r}^\pi(\mathbf{s})$.

Constrained Markov decision process (CMDP) is an augmented version of MDP (Altman, 2021), defined by the tuple $(\mathcal{S}, \mathcal{A}, \mathcal{C}, \mathsf{p}, \mathsf{r})$, in which the set of constraints is defined as: $\mathcal{C} = \left\{ \pi \in \Pi \middle| J_{\mathsf{c}_i}(\pi) \leq d_i, i \in \{1, \ldots, C\} \right\}$, where $J_{\mathsf{c}_i}(\pi) = \mathbb{E}_{\tau \sim \pi}\left[\sum_t \gamma^t \mathsf{c}_i(\mathbf{s}_t, \mathbf{a}_t)\right]$, with $\mathsf{c}_i : \mathcal{S} \times \mathcal{A} \times \mathcal{S} \to \mathbb{R}$. The training objective is then defined as $\max_{\pi \in \mathcal{C}} J_\mathsf{r}(\pi)$, where $\mathcal{C}$ is the constraint (or

feasible) set. In this setting, the corresponding value function, action-value function, and advantage functions for the auxiliary costs are denoted by $V_c^\pi(\mathbf{s})$, $Q_c^\pi(\mathbf{s}, \mathbf{a})$, $A_c^\pi(\mathbf{s}, \mathbf{a})$.

## 3 METHODOLOGY

In this section, we present FALCON, a framework that formulates L2D as a CMDP to address the human-AI cooperation with human performance degradation dependent on workload accumulation. Firstly, we define the human-AI collaborative sequential decision-making task. We then introduce a human performance simulation environment grounded in psychological principles. Lastly, we illustrate the L2D architecture with workload-variant human performance, while introducing constrained optimisation for precise budget control over human-AI cooperation costs.

### 3.1 ENVIRONMENT SETUP

We address sequential classification in the form of episodes. In each episode, a human-AI team collaboratively processes a stream of $T$ sequential data $\tau = \{(\mathbf{x}_t, \mathbf{y}_t)\}_{t=1}^T$, where $\mathbf{x}_t \in \mathcal{X} \subset \mathbb{R}^d$ is an input sample at time step $t$, and $\mathbf{y}_t \in \mathcal{Y} = \{1, \ldots, K\}$ is the corresponding ground truth label. The system maintains two predictive components: 1) a human expert with workload affected performance by defined by $\mathsf{h} : \mathcal{X} \times \mathcal{W} \to \mathcal{Y}$, where $\mathcal{W} \subset \mathbb{R}_+$ is the space that represents the cumulative human workload, and 2) the AI classifier defined by $\mathsf{m} : \mathcal{X} \to \Delta^{K-1}$. At each time step $t$, the system will perform an action $\mathbf{a}_t \in \{\text{AI}, \text{Human}\}$. This action determines which agent will produce the final prediction $\hat{\mathbf{y}}_t$ of the sample $\mathbf{x}_t$.

### 3.2 HUMAN PERFORMANCE SIMULATION

The human performance is simulated with two key assumptions: (1) *Predictable Fatigue Accumulation* (Estes, 2015), where human cognitive performance degrades as a function of cumulative engagement in decision-making tasks, following psychologically grounded fatigue curves ; and (2) *Selective Fatigue* (Hopko et al., 2021), where only tasks assigned to the human expert contribute to fatigue accumulation, while tasks handled by the AI system impose no additional cognitive load.

**Mental Fatigue Curves** Since vigilance wanes as cognitive fatigue accumulates (McCarley & Yamani, 2021; Gyles et al., 2023), we model human performance $\mathsf{w} : \mathcal{W} \to [0, 1]$ using a two-phase piece-wise function:

$$\mathsf{w}(\rho) = \begin{cases} w_0 + (w_{\text{peak}} - w_0) \left( \frac{\rho}{\hat{\rho} \cdot L} \right)^2 & \text{if } 0 \leq \rho \leq \hat{\rho} \cdot L \\ w_{\text{base}} + (w_{\text{peak}} - w_{\text{base}}) \frac{1}{1 + \exp[k(\rho - \bar{\rho} \cdot L)]} & \text{if } \rho \geq \hat{\rho} \cdot L \end{cases}, \tag{2}$$

where $w_0, w_{\text{peak}}, w_{\text{base}}$ denote the initial, peak and minimum (or base) performance levels, $\rho \in \mathcal{W}$ is the cumulative workload (see Eq. (4)), and $\hat{\rho}, \bar{\rho}$ denote the relative workload at the peak performance and at the inflection point of the decay phase, and $k$ is the steepness of performance decline. The function $\mathsf{w}(\rho)$ in Eq. (2) quantifies human performance as it evolves with cumulative workload $\rho$, over a total duration represented by $L$ time steps. The human performance has two distinct phases:

1. *Warm-up* (quadratic growth) (Newell & Rosenbloom, 2013): performance improves from the initial level $w_0$ to peak $w_{\text{peak}}$ as the human adapts to the task.
2. *Fatigue* (sigmoid decay) (Estes, 2015): performance degrades from the peak $w_{\text{peak}}$ toward the minimum $w_{\text{base}}$ due to cognitive fatigue accumulation.

We show three different human performance curves in Fig. 2a by varying the parameters in Eq. (2).

**Human Prediction Modelling** Given the human performance at a particular workload at time step $t$, defined as $\mathsf{w}(\rho_t)$, we model human prediction errors via noise rate $\eta$, which represents the probability at time step $t$ of a classification error, defined as $\eta_t = 1 - \mathsf{w}(\rho_t)$. The prediction distribution of the human prediction given noise rate $\eta_t$ is defined as:

$$\Pr(\hat{\mathbf{y}}|\mathbf{y}, \mathbf{x}, \eta_t) = (1 - \eta_t) \cdot \mathbb{I}(\hat{\mathbf{y}} = \mathbf{y}) + \eta_t/K{-}1 \cdot \mathbb{I}(\hat{\mathbf{y}} \neq \mathbf{y}), \tag{3}$$

where $\mathbf{x}$ and $\mathbf{y}$ denote the data sample and ground truth label, respectively. This means that the human predicts the ground truth label with probability $(1 - \eta_t)$, and one of the $K - 1$ incorrect labels with probability $\eta_t/K{-}1$.

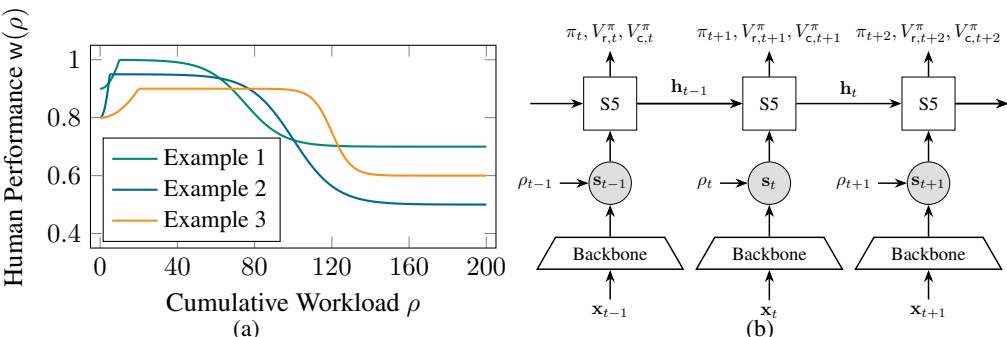

Figure 2: (a): Examples of $\mathsf{w}(\rho)$. The values of parameters $(w_0, w_{\text{peak}}, w_{\text{base}}, k, \bar{\rho}, \hat{\rho})$ in Example 1,2 and 3 are $(0.9, 1, 0.7, 0.1, 0.375, 0.05)$, $(0.8, 0.95, 0.5, 0.09, 0.5, 0.025)$ and $(0.8, 0.9, 0.6, 0.2, 0.6, 0.1)$. (b): The architecture of FALCON with workload-variant human performance. A backbone model extracts visual features from the input $\mathbf{x}_t$, while the cumulative human workload $\rho_t$ is passed through an embedding layer. The visual and workload features are concatenated and processed by a Resettable S5 layers (Lu et al., 2023) to capture temporal dependencies and output the policy $\pi(\mathbf{a}_t|\mathbf{s}_t)$ alongside value estimates.

### 3.3 FATIGUE-AWARE LEARNING TO DEFER VIA CONSTRAINED OPTIMISATION

We model this workflow as a CMDP, where the state space is $\mathbf{s}_t = (\mathbf{x}_t, \rho_t) \in \mathcal{X} \times \mathcal{W}$, where $\mathbf{x}_t$ and $\rho_t$ denote the current input sample and cumulative human workload, respectively. The system transitions deterministically based on the workload update rule:

$$\rho_{t+1} = \begin{cases} \rho_t + 1 & \text{if } \mathbf{a}_t = \text{Human} \\ \rho_t & \text{if } \mathbf{a}_t = \text{AI} \end{cases}, \tag{4}$$

where $\rho_1 = 0$. The reward function is denoted by prediction accuracy (i.e., $\mathsf{r}(\mathbf{s}_t, \mathbf{a}_t) = \mathbb{I}[\hat{\mathbf{y}}_t = \mathbf{y}_t]$), where $\hat{\mathbf{y}}_t$ is the final decision of the system at the time step $t$, while the constraint set $\mathcal{C}$ defines lower and upper limits to human workload, where the lower bound is denoted by $\sum_{t=1}^{T} \mathsf{c}(\mathbf{s}_t, \mathbf{a}_t) \geq d_l$ and the upper limit is defined by $\sum_{t=1}^{T} \mathsf{c}(\mathbf{s}_t, \mathbf{a}_t) \leq d_u$, with $\mathsf{c}(\mathbf{s}_t, \mathbf{a}_t) = \mathbb{I}[\mathbf{a}_t = \text{Human}]$.

**L2D Architecture with workload-variant Human Performance** The architecture of our L2D architecture with workload-variant human performance (Fig. 2b) employs an actor-critic strategy for adaptive decisions. A backbone model takes the input sample $\mathbf{x}_t$ to extract visual feature embeddings, while the cumulative workload $\rho_t$ is embedded by a learnable linear layer. Then the visual and workload features are concatenated and processed through Resettable simplified structured state space sequence (S5) layers (Lu et al., 2023), which represent a variation of structured state space sequence (S4) models (Smith et al., 2023; Gu et al., 2022), to capture temporal dependencies and maintain memory of the human's cognitive state trajectory–this is represented by the state vector $\mathbf{h} \in \mathcal{H}_t \subset \mathbb{R}^H$. From this state vector, three distinct heads predict: the policy $\pi_t = \pi(\mathbf{a}_t|\mathbf{s}_t)$, the estimated future reward $V_{\mathsf{r},t}^\pi = V_{\mathsf{r}}^\pi(\mathbf{s}_t)$, and the estimated future cost $V_{\mathsf{c},t}^\pi = V_{\mathsf{c}}^\pi(\mathbf{s}_t)$.

**Constrained Optimisation with PPO-Lagrangian** We formulate the training phase as a constrained optimisation problem:

$$\max_{\pi_\theta \in \Pi} J_{\mathsf{r}}(\pi_\theta) \qquad \text{s.t.} \quad d_l \leq J_{\mathsf{c}}(\pi_\theta) \leq d_u, \tag{5}$$

where $J_{\mathsf{r}}(\pi_\theta)$ is defined in Section 2.2, $J_{\mathsf{c}}(\pi_\theta) = \mathbb{E}_{\tau \sim \pi_\theta}[\sum_t \gamma^t \mathsf{c}(\mathbf{s}_t, \mathbf{a}_t)]$, with $\mathsf{c}(.)$ defined in equation 4, and $d_l, d_u$ represent the lower and upper limits in cumulated workload. Following the PPO-Lagrangian method (Fujimoto et al., 2019), the constrained problem in equation 5 can be solved via the Lagrangian dual formulation (Altman, 1998):

$$\min_{\lambda_u, \lambda_l \geq 0} \max_{\pi_\theta \in \Pi} J_{\mathsf{r}}(\pi_\theta) - \lambda_u \cdot \max(0, J_{\mathsf{c}}(\pi_\theta) - d_u) - \lambda_l \cdot \max(0, -J_{\mathsf{c}}(\pi_\theta) + d_l). \tag{6}$$

The optimisation of equation 6 involves the update of the Lagrangian multipliers with gradient ascent. If the agent defers too much (exceeding the $d_u$), $\lambda_u$ increases, which heavily penalises the deferral action in the loss function. If the agent defers too little (below the $d_l$), $\lambda_l$ increases, encouraging the deferral action. Note that our constraint objective is to make the cost value between the budget bounds, so we share the same $J_{\mathsf{c}}(\pi_\theta)$ with the two Lagrange multipliers. For instance, if we want 0.4 target coverage, we set $d_u = 0.65$ and $d_l = 0.55$. Please refer to Appendix A.2 for the updates of PPO-Lagrangian, while Algorithms 1 and 2 for training and testing procedures of FALCON.

## 4    FATIGUE-AWARE L2D (FA-L2D) BENCHMARK

Our new benchmark is designed to evaluate L2D methods under the assumption that humans have a variable performance as a function of cumulative workload. During each training episode, images are randomly sampled from the training set, while human performance parameters are randomly sampled from predefined ranges (See Tables 4 to 7 for dataset-specific human performance parameter ranges). As fatigue accumulates according to Eq. (2), human predictions are modelled probabilistically with a noise rate $\eta_t = 1 - \mathrm{w}(\rho_t)$, while humans predict correctly with probability $1 - \eta_t$ and make random classification errors among the remaining $K - 1$ classes with probability $\eta_t/K-1$. This design enables the generation of diverse scenarios ranging from a nearly static expert performance to highly variable fatigue patterns, providing a more realistic representation of human-AI cooperation environments than previous benchmarks that assumed temporal stability. The controllable nature of fatigue parameters allows systematic evaluation of L2D methods across different human performance profiles, from minimally fatigued experts to those experiencing significant cognitive decline over time.

**Datasets**    *Cifar100* (Krizhevsky & Hinton, 2009) has 50k training images and 10k testing images, with each image belonging to one of 100 classes. *Chaoyang* (Zhu et al., 2021) comprises 6,160 colon slide patches categorised into four classes. *MiceBone* (Schmarje et al., 2022) has 7,240 second-harmonic generation microscopy images, where the annotation consists of one of three possible classes. *FLickr10K* (Yang et al., 2017) is a large-scale dataset containing 10,700 images labelled with 8 commonly used emotions. To ensure fair comparison across all methods, we standardise the testing episodes by reshuffling several datasets. Please refer to Appendix C for the datasets details.

**Metrics**    We evaluate performance using prediction accuracy as a function of coverage on test set episodes, where coverage denotes the percentage of samples classified solely by the AI. These accuracy–coverage curves capture the trade-off between accuracy and cooperation budget as coverage varies from 0% (human-only classification) to 100% (AI-only classification). Reported results are averaged over three models trained with different random seeds and evaluated at the final training epoch. To provide a concise quantitative summary, we compute the *area under the accuracy–coverage curve* (AUACC), where higher AUACC values indicate more favourable accuracy–coverage trade-offs.

**Ablation Settings**    To systematically evaluate L2D methods across different human performance patterns, we define three distinct benchmark cases that capture varying degrees of fatigue-induced performance degradation as follows:

1. *Sustained High Performance:* This case models scenarios where human experts maintain consistently high performance throughout the task duration, approximating the static expert assumption used in traditional L2D methods.

2. *Normal Fatigue:* This case represents typical workplace conditions where human performance follows a standard warm-up and fatigue cycle.

3. *Rapid Fatigue:* This case simulates sharp decline in human performance, aiming to test the robustness of L2D methods under extreme cognitive fatigue conditions, such as those encountered during extended work shifts or high-stress environments.

For each case, we employ two distinct evaluation protocols to assess robustness and generalisation capability. **Fine-tuning Setting:** This setting evaluates how well different approaches can adapt when they have full knowledge of the specific fatigue pattern during training. All training procedures remain consistent with the main experiments, but the human performance simulation for training and testing uses only the parameters from the specific case being evaluated, rather than the broader parameter ranges shown in Fig. 7. **Zero-shot Setting:** This setting measures the ability to generalise to previously unseen human performance patterns. Methods use models trained on the main experiments with parameter ranges in Tables 4 to 7 and Fig. 7 without additional training or adaptation for the specific case being evaluated during testing.

## 5    EXPERIMENTS

For our training, Cifar100 has 200 steps in each episode, while the other datasets have 100 steps[1] The training parameters for PPO and the Lagrange multipliers are in Table 3, while the parameters of mental fatigue curves $\mathrm{w}(\rho)$ are provided in Tables 4 to 7 for different datasets in Appendix.

---

[1]Cifar100 uses a larger number of episodes because it has more images than other datasets in FA-L2D.

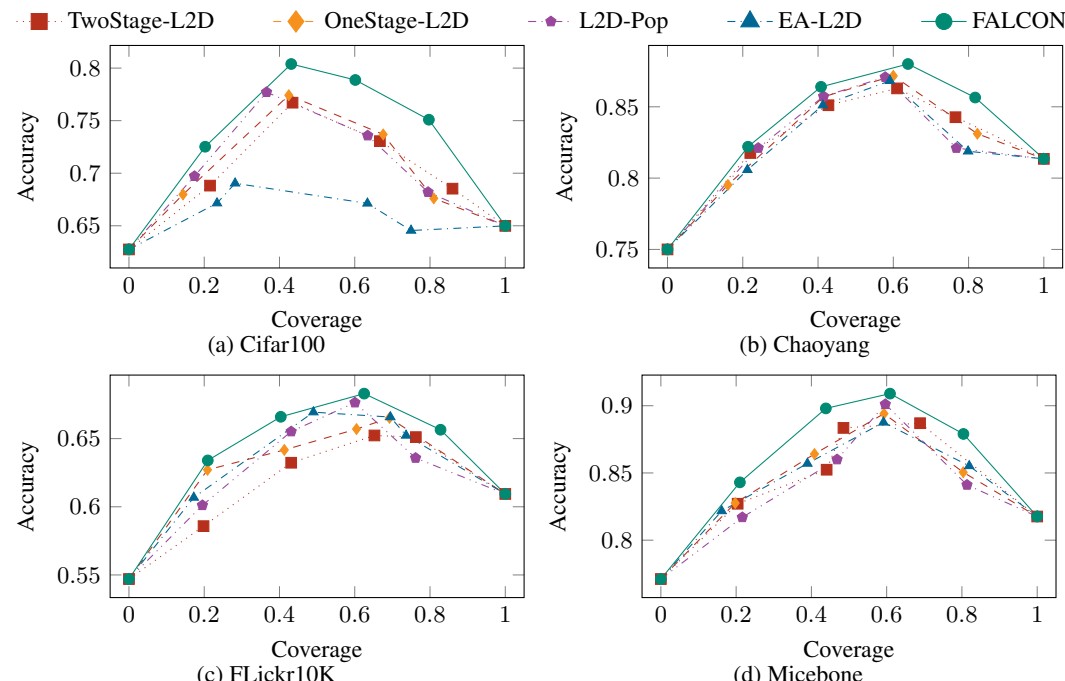

Figure 3: Accuracy-Coverage curves of several L2D strategies and FALCON on various datasets.

**Baselines and SOTAs** We compare FALCON with SOTA L2D methods, such as one-stage L2D (Mozannar & Sontag, 2020a), two-stage L2D (Madras et al., 2018), L2D-Pop (Tailor et al., 2024), and EA-L2D (Strong et al., 2025). During the training phase, conventional L2D methods require all human predictions for backpropagation. Therefore, to train these models, we simulate human experts by randomly generating a complete sequence of performance variations for each training trajectory. For L2D-Pop and EA-L2D, we follow its prescribed methodology by training it on 16 randomly generated human expert simulations per epoch. In the testing phase, after the model makes its deferral decisions, the accuracy is calculated by incorporating the simulated human's predictions for all deferred instances. Note that we control the budget of all static L2D methods by the penalty constraint optimisation from (Zhang et al., 2024). Please refer to Appendix B for details.

**Comparison with Baselines and SOTAs** We report the *accuracy-coverage curves* of several L2D strategies and our proposed FALCON across the FA-L2D benchmark datasets in Fig. 3. In general, FALCON outperforms all competing methods at every coverage level in all benchmarks. TwoStage-L2D achieves better performance at high coverage but worse than others at low coverages. On datasets with a small number of classes, (e.g. Chaoyang, Micebone), L2D-Pop and EA-L2D show small improvements over simpler OneStage and TwoStage L2D models. This suggests that their learned human representation is weak in these scenarios. In contrast, FALCON maintains a remarkable performance advantage, especially when coverage is large, highlighting its capabilities. On the FLickr10K dataset, L2D-Pop and EA-L2D outperform the simpler baselines at mid-range budget levels. This indicates that they can capture an average representation of expert performance. However, FALCON's strength lies in its ability to adapt to a dynamic environment and unseen expert behaviours, rather than relying on a simple average. EA-L2D performs worse than other methods on datasets with a large number of classes (e.g., Cifar100), because the counting-based prior for expert accuracy cannot scale effectively. When the number of classes increases, the gating function will be biased to the classifier. Although L2D-Pop achieves higher performance than other baselines, FALCON achieves the best results. Regarding the AUACC results in Table 1 of Appendix, FALCON shows better results than all other methods for all datasets. It is worth noting the superior performances, particularly on Cifar100 and MiceBone. All other methods perform competitively against each other, except for EA-L2D that shows poor performance on Cifar100.

**Robustness of L2D Methods Under FA-L2D Parameter Variations** To evaluate the robustness of L2D methods to varying parameters of the FA-L2D benchmark, we test the methods with the ablation settings in Section 4, as illustrated in Figs. 4a, 4d and 4g (above) and Table 2 (in Appendix). We

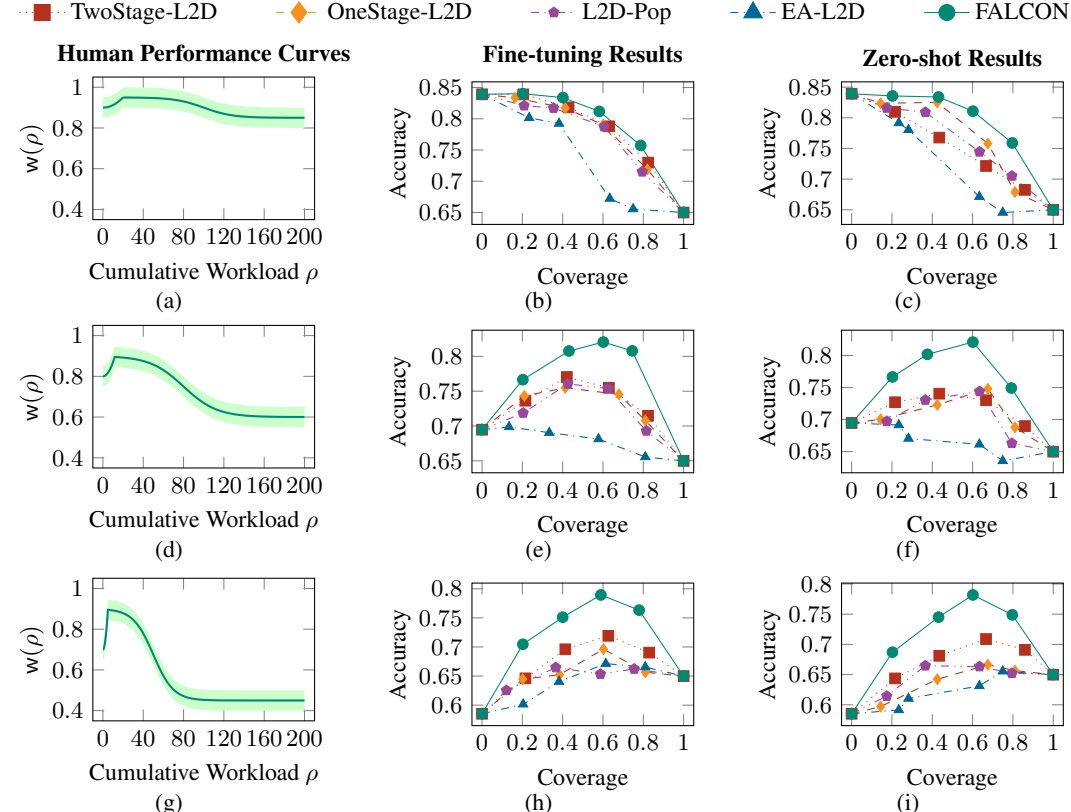

Figure 4: Different human performance during testing (left column) and corresponding results with fine-tuning (middle column) and zero-shot (right column) testing on Cifar100 across all methods.

evaluated each method's ability to adjust its deferral strategy under fine-tuning settings in Figs. 4b, 4e and 4h and zero-shot settings in Figs. 4c, 4f and 4i.

*Sustained High Performance.* In this scenario, the human expert's accuracy remains high, staying above 80% for the duration of the task in Fig. 4a. The results in Figs. 4b and 4c show that FALCON consistently achieves the highest accuracy. Other methods achieve similar performance in Fig. 4a, which indicates their advantages in standard L2D setting. In Fig. 4c, FALCON significantly exceeds that of EA-L2D and L2D-Pop, which struggle to effectively cooperate with a strong human expert. OneStage-L2D performs better than EA-L2D and L2D-Pop, suggesting that these methods cannot learn efficient dynamic human presentation.

*Normal Fatigue.* In this case, human performance peaks around 40 steps before gradually decreasing in Fig. 4d. By letting the model learn the relatively slow human performance variation, performance in Fig. 4e is better than in Fig. 4f, but also lag behind FALCON. In Fig. 4f, the performance of all the methods is close to the results in Fig. 3a.

*Rapid Fatigue.* Human performance decreases from above 90% to below 50% in the first 80 time steps (Fig. 4g). This challenging condition highlights the robustness of our approach. The results in Figs. 4h and 4i show that FALCON maintains high precision by correctly identifying human unreliability and adjusting its deferral strategy. In contrast, all other methods suffer a significant performance drop. While TwoStage-L2D is the best of the baselines, its accuracy still lags behind FALCON at all coverage levels.

**Remark 1** *In the almost static L2D setting (e.g., sustained high-performance humans), it becomes harder to appreciate the value of L2D methods (see Fig. 4a and results in Figs. 4b and 4c): L2D does not exhibit improved performance for coverages strictly between 0 and 1. By contrast, in the normal and rapid fatigue settings (Figs. 4d and 4g), L2D methods surpass both AI-only and human-only baselines across intermediate coverage levels (Figs. 4e, 4f, 4h and 4i), highlighting the effectiveness of adaptive human–AI cooperation in scenarios that more closely mirror real-world conditions.*

## 6 RELATED WORK

**Learning to defer** aims to learn a classifier and a rejector to decide in which case the decision should be deferred to a human expert to make the final prediction (Madras et al., 2018). Existing L2D approaches focus on the development of different surrogate loss functions to be consistent with the Bayes-optimal classifier (Mozannar & Sontag, 2020a; Wei et al., 2024; Cao et al., 2024). However, these methods overlook the settings that contain a wide diversity of multiple human experts. Recently, research in L2D shifts towards the multiple-expert setting (Mao et al., 2023; Verma et al., 2023; Mao et al., 2024; Zhang et al., 2025; Nguyen et al., 2025), and unseen expert scenarios (Tailor et al., 2024; Strong et al., 2025). For example, L2D-Pop (Tailor et al., 2024) proposed to encode image and corresponding human labels into a conditional latent context set for human representations. During testing, given few-shot context predictions of experts, the gating model can use fine-grained human embeddings to make deferral decisions. EA-L2D (Strong et al., 2025) constructs an explicit Bayesian representation of expert performance from their context predictions by counting correct predictions of each expert. Sequential Learning-to-Defer (SLTD) (Joshi et al., 2023) frames the L2D problem as a model-based reinforcement learning task, which considers the causal chain of events in the environment. Different from our method, SLTD focuses on scenarios where the underlying rules or data distribution of the task change over time. Furthermore, SLTD assumes access to prior batch data collected by human experts, which is expensive for L2D systems.

**Human mental fatigue** is a critical component of non-technical skills within human factors research (Casali et al., 2019). Mental fatigue is a psychobiological state resulting from prolonged cognitive engagement (Driskell & Salas, 2013; Van Cutsem et al., 2022). This phenomenon manifests across multiple dimensions: physiologically through measurable changes in brain activity (Müller et al., 2021), behaviourally through systematic declines in cognitive performance (Lindner & Retelsdorf, 2020), and subjectively through increased perception of effort and diminished energy (Hockey, 2013). Recently, research suggested that mental fatigue can affect physical performance (Enoka & Duchateau, 2016; Van Cutsem et al., 2017; Marcora et al., 2009; Dallaway et al., 2022). The temporal dynamics of fatigue-induced performance degradation follow distinct mathematical patterns that depend on task characteristics. For simple repetitive or vigilance tasks, performance typically follows exponential decay curves (Anderson, 2013). Jaber et al. (2013) proposed exponential models to capture fatigue and recovery cycles across repeated work-rest periods. However, complex adaptive tasks requiring sustained cognitive engagement exhibit sigmoid performance patterns (Enoka & Duchateau, 2016; Gyles et al., 2023). Leppink & Pérez-Fuster (2019) observed that mental effort scales non-linearly with workload and time, with vigilance decline following non-linear patterns (McCarley & Yamani, 2021) and cognitive load relationships exhibiting sigmoid curves (Estes, 2015).

Unlike existing L2D methods that treat human experts as static oracles, our approach explicitly models workload-dependent performance degradation using psychologically-grounded functions, enabling more informed deferral decisions by accounting for the expert's current cognitive state rather than assuming constant capability throughout task sequences.

## 7 CONCLUSION

In this paper, we proposed FALCON to explicitly model dynamic human performance degradation due to cognitive fatigue. By formulating L2D as a CMDP with psychologically-grounded fatigue curves and PPO-Lagrangian optimisation, FALCON addresses the unrealistic assumption of static human expert performance in existing methods. Extensive experiments on our proposed FA-L2D benchmark demonstrated that FALCON consistently outperformed SOTA L2D approaches across all coverage levels and achieved robust zero-shot generalisation to unseen expert fatigue patterns.

While FALCON captures general patterns of cognitive decline, it may not fully represent individual variation in fatigue patterns across different populations or task contexts. Also, the current evaluation relies on simulated human performance rather than real human studies. FALCON assumes uniform fatigue contribution across tasks, but cognitive load varies significantly with task complexity. In future work, we will incorporate instance-dependent fatigue modelling, extending Eqs. (3) and (4) to account for instance-specific cognitive load, while incorporating multi-modal sensitive fatigue indicators and real-world deployment studies with real human experts.

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

# A  APPENDIX

## A.1  DISCLOSURE OF LLM USAGE

Large language models were used to improve grammar and assist with paper revision. All ideas, experimental design, data analysis, and conclusions are the original work of the authors.

## A.2  PPO-LAGRANGIAN OPTIMISATION

The critic model is optimised by regression on mean-square error between value estimator and the true trajectory value, followed by the standard PPO setting. The policy update follows standard PPO-Lagrangian with modified objective:

$$J_{\mathsf{r}}(\pi_{\boldsymbol{\theta}}) = \mathbb{E}_{\tau \sim \pi_{\boldsymbol{\theta}_{old}}} \left[ \min \left( \frac{\pi_{\boldsymbol{\theta}}(\mathbf{a}|\mathbf{s})}{\pi_{\boldsymbol{\theta}_{old}}(\mathbf{a}|\mathbf{s})} A_{\mathsf{r}}^{\pi_{\boldsymbol{\theta}_{old}}}(\mathbf{s}, \mathbf{a}), \mathrm{clip} \left( \frac{\pi_{\boldsymbol{\theta}}(\mathbf{a}|\mathbf{s})}{\pi_{\boldsymbol{\theta}_{old}}(\mathbf{a}|\mathbf{s})}, 1 - \epsilon, 1 + \epsilon \right) A_{\mathsf{r}}^{\pi_{\boldsymbol{\theta}_{old}}}(\mathbf{s}, \mathbf{a}) \right) \right] \tag{7}$$

$$J_{\mathsf{c}}(\pi_{\boldsymbol{\theta}}) = \mathbb{E}_{\tau \sim \pi_{\boldsymbol{\theta}_{old}}} \left[ \min \left( \frac{\pi_{\boldsymbol{\theta}}(\mathbf{a}|\mathbf{s})}{\pi_{\boldsymbol{\theta}_{old}}(\mathbf{a}|\mathbf{s})} A_{\mathsf{c}}^{\pi_{\boldsymbol{\theta}_{old}}}(\mathbf{s}, \mathbf{a}), \mathrm{clip} \left( \frac{\pi_{\boldsymbol{\theta}}(\mathbf{a}|\mathbf{s})}{\pi_{\boldsymbol{\theta}_{old}}(\mathbf{a}|\mathbf{s})}, 1 - \epsilon, 1 + \epsilon \right) A_{\mathsf{c}}^{\pi_{\boldsymbol{\theta}_{old}}}(\mathbf{s}, \mathbf{a}) \right) \right] \tag{8}$$

where $\boldsymbol{\theta}_{old}$ is the vector of policy parameters before the update, clip(.) is the clipping operation on the probability ratio with clipping parameter $\epsilon$, which controls the maximum allowed deviation of the updated policy from the previous policy to ensure stable training.

Furthermore, we update the Lagrangian multipliers using gradient ascent with an Adam optimiser:

$$\lambda_u^{(\beta+1)} = \max(0, \lambda_u^{(\beta)} + \alpha_\lambda \cdot (J_{\mathsf{c}}(\pi) - d_u)) \tag{9}$$

$$\lambda_l^{(\beta+1)} = \max(0, \lambda_l^{(\beta)} + \alpha_\lambda \cdot (d_l - J_{\mathsf{c}}(\pi))) \tag{10}$$

where $\lambda_u$ and $\lambda_l$ are the Lagrangian multipliers for the upper and lower bound constraints respectively, $\alpha_\lambda$ is the learning rate for the multiplier updates, $d_u$ and $d_l$ denote the upper and lower constraint thresholds, $\beta$ denotes the update step, and $J_{\mathsf{c}}(\pi)$ represents the expected cumulative cost under policy $\pi$. The multipliers automatically adjust to enforce the constraint bounds: $\lambda_u$ increases when the cost exceeds the upper limit $d_u$, penalizing excessive human utilisation, while $\lambda_l$ increases when the cost falls below the lower limit $d_l$, encouraging sufficient human engagement.

## A.3  DESIGN CHOICE OF S5 FOR FALCON

Modelling human cognitive state over long episodes is critical for FALCON to track cumulative fatigue, making the choice of sequence model critical. RL systems typically employ RNNs (Yan et al., 2023; Jha et al., 2025; Gessler et al., 2025; Morad et al., 2023; David et al., 2022; Lu et al., 2023), Transformers (Chen et al., 2021; Parisotto et al., 2020), and as the sequence models. However, traditional RNNs, such as LSTM and GRU, suffer vanishing gradients over long sequences while Transformers incur quadratic computational costs prohibitive for extended episodes (Metz et al., 2021) We employ Resettable Simplified Structured State Space Sequence (S5) layers (Lu et al., 2023), a variant of S4 models (Smith et al., 2023; Gu et al., 2022), which provide linear computational complexity and stable gradient flow essential for tracking cumulative workload over hundreds of time steps. S5 demonstrates superior asymptotic runtime compared to Transformers while significantly outperforming LSTMs in both performance and computational efficiency (Lu et al., 2023).

# B  IMPLEMENTATION DETAILS

## B.1  ARCHITECTURE

All methods are implemented in Jax, a Python library that accelerates array computation and program transformation to achieve high-performance numerical computing for large-scale machine learning,

Table 1: Quantitative comparison in terms of AUACC (×100) (Nadeem et al., 2009) of the SOTA L2D (Mozannar & Sontag, 2020a; Madras et al., 2018; Tailor et al., 2024; Strong et al., 2025) on the L2D datasets. The results consist of the mean and standard deviations obtained from three experiments using models trained with different random seeds. The best result per benchmark is marked in bold.

|  | Cifar100 | Chaoyang | FLickr10K | MiceBone |
|---|---|---|---|---|
| OneStage L2D | 70.87±0.13 | 83.24±0.14 | 63.06±0.12 | 84.61±0.12 |
| TwoStage L2D | 70.50±0.15 | 83.15±0.08 | 61.77±0.13 | 84.58±0.15 |
| L2D-Pop | 71.01±0.11 | 82.65±0.17 | 62.72±0.18 | 83.96±0.14 |
| EA-L2D | 66.26±0.39 | 82.39±0.08 | 63.26±0.23 | 84.59±0.12 |
| Ours | **74.01±0.09** | **84.13±0.11** | **64.40±0.08** | **86.08±0.13** |

Table 2: Quantitative comparison in terms of the Area Under Accuracy-Coverage Curve (AUACC) (×100) (Nadeem et al., 2009) of the SOTA L2D (Mozannar & Sontag, 2020a; Madras et al., 2018; Tailor et al., 2024; Strong et al., 2025) with three different human performance curves on the Cifar100 dataset. The results consist of the mean value obtained from three experiments using models trained with different random seeds. The best result per benchmark is marked in bold.

|  | Sustained High Performance | | Normal Fatigue | | Rapid Fatigue | |
|---|---|---|---|---|---|---|
|  | Fine-tuning | Zero-shot | Fine-tuning | Zero-shot | Fine-tuning | Zero-shot |
| OneStage L2D | 78.23±0.17 | 77.25±0.13 | 72.67±0.10 | 70.83±0.16 | 66.75±0.09 | 63.85±0.11 |
| TwoStage L2D | 78.78±0.14 | 75.22±0.16 | 73.10±0.09 | 71.56±0.07 | 67.36±0.14 | 67.49±0.13 |
| L2D-Pop | 77.63±0.08 | 76.38±0.12 | 72.07±0.11 | 70.35±0.13 | 64.82±0.12 | 64.40±0.08 |
| EA-L2D | 73.46±0.10 | 72.23±0.14 | 67.87±0.15 | 66.51±0.12 | 63.97±0.17 | 62.31±0.14 |
| Ours | **79.58±0.10** | **79.70±0.12** | **76.93±0.07** | **76.20±0.09** | **72.36±0.15** | **71.68±0.07** |

while running on a single Nvidia RTX A6000. A mixed precision using `bfloat16` is applied over all methods and datasets to speed up the training. All AI models are trained for 300 epochs using stochastic gradient descent with a momentum of 0.9 and a learning rate of 0.01. The learning rate is decayed through a cosine decaying scheduler, and the gradient norm is clipped at the maximal of 10 for numerical stability. For experiments performed on Cifar100 dataset, we employ PreAct-ResNet-18 and the batch size used is 256. For other datasets, we train the AI model with a ResNet-18 using a regular CE loss minimisation with a ground truth label, while the batch size used is 256. On Cifar100 the AI model achieves 64.99% accuracy on the testing set. The AI models on Chaoyang, Flickr10K, and Micebone datasets achieve 72.65%, 81.35%, 60.94%, and 81.76%, respectively. Furthermore, the actor, reward and cost, critic heads in Fig. 2b consist of two-layer multi-layer perceptron (MLP), where each hidden layer has 512 nodes activated by Rectified Linear Units (ReLU).

### B.2 GATING MODEL TRAINING

All methods are trained for 1e7 iterations. For our PPO-Lagrangian training, we use Adam optimiser and the parameters is shown in Table 3. For other methods, we employ stochastic gradient descent with a momentum of 0.9, while the initial learning rate is set at 0.01 and decayed through a cosine annealing.

### B.3 INFERENCE TIME

The inference time for 50 episodes on Cifar100 dataset across all methods in shown in Fig. 6, FALCON showcases similar inference time compared to other static L2D methods.

## C DATASETS

**Cifar100** (Krizhevsky & Hinton, 2009) has 50k training images and 10k testing images, with each image belonging to one of 100 classes categorised into 20 super-classes. In addition, because about 10% of testing images in Cifar100 (Krizhevsky & Hinton, 2009) are duplicated or almost identical to

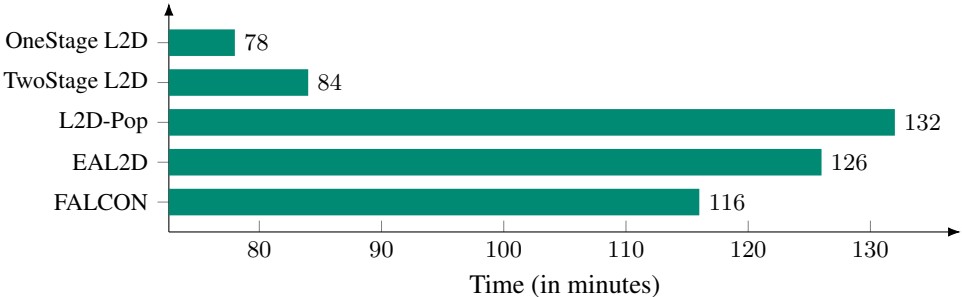

Figure 5: Training time of FALCON and competing methods on Cifar100 (1e7 iterations).

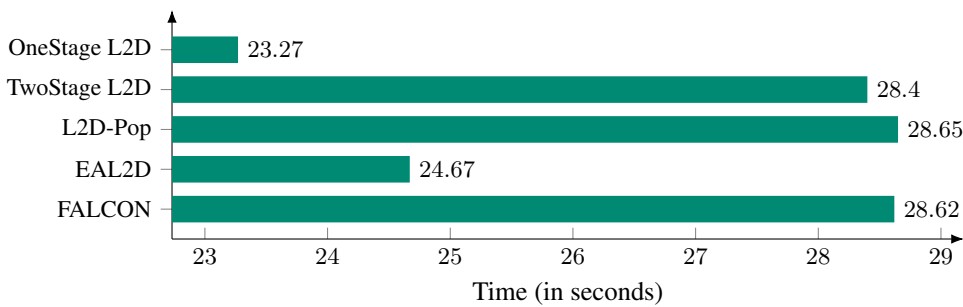

Figure 6: Inference time of FALCON and competing methods on Cifar100 (50 episodes).

the ones in the training set, in our training and testing, we use ciFAIR-100 (Barz & Denzler, 2020), which replaces those duplicated images by different images belonging to the same class.

**Chaoyang** (Zhu et al., 2021) comprises 6,160 colon slide patches categorised into four classes: *normal, serrated, adenocarcinoma, and adenoma*, where each patch has *three noisy labels annotated by three pathologists*. In the original Chaoyang dataset setup, the training set has patches with multi-rater noisy labels, while the testing set only contains patches that all experts agree on a single label. We assume that the majority vote forms the ground truth annotation.

**MiceBone** (Schmarje et al., 2022) has 7,240 second-harmonic generation microscopy images, with each image being annotated by one to five professional annotators, where the annotation consists of one of three possible classes: *similar collagen fiber orientation, dissimilar collagen fiber orientation, and not of interest due to noise or background*. Only 8 out of 79 annotators label the whole dataset. We, therefore, use the majority vote of 8 annotators as the ground truth.

**Flickr10K** (Yang et al., 2017) is a subset of Flickr dataset (Borth et al., 2013), in which the numbers of each class are roughly equal. It contains 10,700 images labelled with 8 commonly used emotions, including *amusement, contentment, excitement, awe, anger, disgust, fear, and sadness*.

**Dataset Reshuffling.** To ensure fair comparison across all methods, we standardise the testing episodes by reshuffling several datasets. For **Cifar100**, we retain the original test set, resulting in 50 testing episodes with 200 time steps each. For **Chaoyang**, we split the complete dataset into 4,160 training images and 2,000 testing images, yielding 20 testing episodes with 100 time steps. For **MiceBone**, we allocate 5,240 images for training and reserve the remaining 2,000 images for testing, comprising 20 episodes with 100 time steps. For **FLickr10K**, we divide the dataset into 8,700 training images and 2,000 testing images, which generates 20 testing episodes with 100 time steps.

---

**Algorithm 1** FALCON training procedure

---

1: **procedure** TRAINING($\mathcal{D}, n_{\text{iter}}, n_{\text{episode}}$)
2:  ▷ $\mathcal{D} = \{\mathbf{x}_t, \mathbf{y}_t\}_{t=1}^T$: *training dataset*  ◁
3:  ▷ $n_{iter}$: *the total number of iterations*  ◁
4:  ▷ $n_{episode}$: *number of episodes*  ◁
5:  initialise AI classifier m, policy $\pi_{\boldsymbol{\theta}_1}$, value function $V_{\mathsf{r}}^{\boldsymbol{\phi}_1}$ and cost value function $V_{\mathsf{c}}^{\boldsymbol{\psi}_1}$
6:  initialise Lagrangian multiplier $\lambda_u, \lambda_l$
7:  **for** $j = 1$ to $n_{\text{iter}}$ **do**
8:   collect set of trajectories: $\hat{\mathcal{D}}_j \leftarrow$ COLLECT TRAJECTORIES($\mathcal{D}, \mathsf{m}, \pi_{\boldsymbol{\theta}}, V_{\mathsf{r}}^{\boldsymbol{\phi}}, V_{\mathsf{c}}^{\boldsymbol{\psi}}, n_{\text{episode}}$)
9:   update Lagrangian multiplier $\lambda_u, \lambda_l$ via gradient ascent  ▷ *defined in Eq.* (10)
10:   compute estimated reward value $\hat{\mathsf{r}}_t = \sum_{j=0}^{T-t} \gamma^j \mathsf{r}_{t+j}$ and reward advantage $A_{\mathsf{r}}^{\pi_{\boldsymbol{\theta}_i}}$
11:   compute estimated cost value $\hat{\mathsf{c}}_t = \sum_{j=0}^{T-t} \gamma^j \mathsf{c}_{t+j}$ and cost advantage $A_{\mathsf{c}}^{\pi_{\boldsymbol{\theta}_j}}$
12:   shuffle data in $\hat{\mathcal{D}}_j$ and split into mini-batches
13:   **for** each mini-batch from $\hat{\mathcal{D}}_j$ **do**
14:    update $\pi_{\boldsymbol{\theta}_{j+1}}$ using PPO  ▷ *defined in Eq.* (8)
15:    update reward value function: $\boldsymbol{\phi}_{j+1} \leftarrow \arg\min_{\boldsymbol{\phi}} \frac{1}{|\hat{\mathcal{D}}_j|T} \sum (V_{\mathsf{r}}^{\boldsymbol{\phi}_j} - \hat{\mathsf{r}}_t)^2$
16:    update cost value function: $\boldsymbol{\psi}_{j+1} \leftarrow \arg\min_{\boldsymbol{\psi}} \frac{1}{|\hat{\mathcal{D}}_j|T} \sum (V_{\mathsf{c}}^{\boldsymbol{\psi}_j} - \hat{\mathsf{c}}_t)^2$
17:  **return** the optimal policy $\boldsymbol{\theta}_{n_{iter}}$

18: **procedure** COLLECT TRAJECTORIES($\mathcal{D}, \mathsf{m}, \pi_{\boldsymbol{\theta}}, V_{\mathsf{r}}^{\boldsymbol{\phi}}, V_{\mathsf{c}}^{\boldsymbol{\psi}}, n_{\text{episode}}$)
19:  ▷ $\mathcal{D}$: *training dataset*  ◁
20:  ▷ m: *AI classifier*  ◁
21:  ▷ $\pi_{\boldsymbol{\theta}}$: *policy function parameterised by* $\boldsymbol{\theta}$  ◁
22:  ▷ $V_{\mathsf{r}}^{\boldsymbol{\phi}}$: *reward value function parameterised by* $\phi$  ◁
23:  ▷ $V_{\mathsf{c}}^{\boldsymbol{\psi}}$: *cost value function parameterised by* $\psi$  ◁
24:  ▷ $n_{episode}$: *number of episodes*  ◁
25:  set data buffer $\hat{\mathcal{D}} = \varnothing$
26:  **for** $i = 1$ to $n_{\text{episode}}$ **do**
27:   sample a sequences of $T$ images from $\mathcal{D}$
28:   sample fatigue model parameters $w_0, w_{peak}, w_{base}, k, \bar{\rho}, \hat{\rho}$  ▷ *See Tables 4 to 7*
29:   initialise human workload accumulator $\rho \leftarrow 0$ and $\mathsf{w}(0) \leftarrow w_0$
30:   **for** $t = 1$ to $T$ **do**
31:    get current state: $\mathbf{s}_t \leftarrow (\mathbf{x}_t, \rho)$
32:    sample an action from the policy: $\mathbf{a}_t \sim \pi_{\boldsymbol{\theta}_i}(\mathbf{s}_t)$
33:    **if** $\mathbf{a}_t =$ human **then**  ▷ *human expert makes the prediction*
34:     update human workload: $\rho \leftarrow \rho + 1$
35:     update human performance: $\mathsf{w}_t \leftarrow \mathsf{w}(\rho)$  ▷ *defined in Eq.* (2)
36:     get the annotation flipping probability of human due to fatigue: $\eta \leftarrow 1 - \mathsf{w}_t$
37:     sample human prediction: $\hat{\mathbf{y}}_t \sim \Pr(\hat{\mathbf{y}}_t | \mathbf{y}_t, \eta)$  ▷ *defined in Eq.* (3)
38:    **else if** $\mathbf{a}_t =$ AI **then**  ▷ *AI classifier makes the prediction*
39:     get the label predicted by the classifier: $\hat{\mathbf{y}}_t \leftarrow \arg\max \mathsf{m}(\mathbf{x}_t)$
40:    $\mathsf{r}_t \leftarrow \mathbb{I}(\mathbf{y}_t = \hat{\mathbf{y}}_t)$
41:    gather data from $\pi(\cdot|\mathbf{s}_t, \mathbf{a}_t)$, then $\hat{\mathcal{D}} = \hat{\mathcal{D}} \cup \{\tau_{t+1}, \mathbf{s}_t, \mathbf{a}_t, \mathsf{r}_t, \log \pi_{\boldsymbol{\theta}}(\mathbf{a}_t|\mathbf{s}_t), V_{\mathsf{r}}^{\boldsymbol{\phi}}, V_{\mathsf{c}}^{\boldsymbol{\psi}}\}$
42:  **return** $\hat{\mathcal{D}}$

**Algorithm 2** FALCON testing procedure

1: **procedure** TESTING($\mathcal{D}$, w, m, $\boldsymbol{\theta}$)
2:     ▷ $\mathcal{D} = \{\mathbf{x}_t, \mathbf{y}_t\}_{t=1}^{T}$: *testing dataset*     ◁
3:     ▷ $\boldsymbol{\theta}$: *parameter of policy function*     ◁
4:     ▷ w: *fatigue function*     ◁
5:     ▷ m: *AI classifier*     ◁
6:     sample fatigue model parameters $w_0, w_{peak}, w_{base}, k, \bar{\rho}, \hat{\rho}$     ▷ *See Tables 4 to 7*
7:     initialise human workload accumulator $\rho \leftarrow 0$ and $w(0) \leftarrow w_0$
8:     initialise accumulate accuracy r
9:     **for** $t = 1$ to $T$ **do**
10:        get current state: $\mathbf{s}_t \leftarrow (\mathbf{x}_t, \rho)$
11:        select an action: $\mathbf{a}_t \leftarrow \arg\max_{\boldsymbol{\theta}} \pi_{\boldsymbol{\theta}}(\mathbf{s}_t)$
12:        **if** $\mathbf{a}_t =$ human **then**     ▷ *human expert makes the prediction*
13:           update human workload: $\rho \leftarrow \rho + 1$
14:           update human performance: $w_t \leftarrow w(\rho)$     ▷ *defined in Eq. (2)*
15:           get the annotation flipping probability of human due to fatigue: $\eta \leftarrow 1 - w_t$
16:           sample human prediction: $\hat{\mathbf{y}}_t \sim \Pr(\hat{\mathbf{y}}_t | \mathbf{y}_t, \eta)$     ▷ *defined in Eq. (3)*
17:        **else if** $\mathbf{a}_t =$ AI **then**     ▷ *AI classifier makes the prediction*
18:           get the label predicted by the classifier: $\hat{\mathbf{y}}_t \leftarrow \arg\max \mathsf{m}(\mathbf{x}_t)$

19:        ▷ *Calculate accuracy* $r_t$     ◁
20:        **if** $\hat{\mathbf{y}}_t = \mathbf{y}_t$ **then**
21:           $r_t \leftarrow 1$     ▷ *correct prediction*
22:        **else**
23:           $r_t \leftarrow 0$     ▷ *incorrect prediction*
24:        $r \leftarrow \hat{r} + r_t$     ▷ *accumulate reward*
25:     **return** $r/T, 1 - \rho/T$     ▷ *return accuracy and coverage*

Table 3: PPO parameters

| Params | Value |
| --- | --- |
| Activation | Relu |
| Clipping_Coefficient $\epsilon$ | 0.2 |
| Entropy_Coefficient | 0.001 |
| Lagrangian_LR | 0.035 |
| Lagrangian_INIT $\lambda$ | 0.001 |
| GAE_LAMBDA | 0.95 |
| Discount Factor $\gamma$ | 0.99 |
| LR | 0.0004 |
| LR_WARMUP | 0.01 |
| UPDATE_EPOCHS | 4 |
| Value Function Weight | 0.5 |
| Maximum Gradient Norm | 0.5 |
| S5 Layers | 4 |
| S5 Hidden Size | 512 |
| FC_DIM | 512 |

Table 4: The range of parameters of human performance variation in Eq. (2) on Cifar100 dataset.

| Params | Range | Description |
| --- | --- | --- |
| $w_0$ | $\mathcal{U}(0.7, 0.9)$ | initial performance |
| $w_{\text{base}}$ | $\mathcal{U}(0.4, 0.5)$ | minimum performance |
| $w_{\text{peak}}$ | $\mathcal{U}(0.8, 1.0)$ | maximum performance |
| $\hat{\rho}$ | $\mathcal{U}(0.025, 0.1)$ | relative workload at the peak performance |
| $\bar{\rho}$ | $\mathcal{U}(0.25, 0.5)$ | relative workload at the inflection point of the decay phase |
| $k$ | $\mathcal{U}(0.05, 0.1)$ | steepness of performance decline |

Table 5: The range of parameters of human performance variation in Eq. (2) on Chaoyang dataset.

| Params | Range | Description |
|---|---|---|
| $w_0$ | $\mathcal{U}(0.8, 0.9)$ | initial performance |
| $w_{\text{base}}$ | $\mathcal{U}(0.6, 0.7)$ | minimum performance |
| $w_{\text{peak}}$ | $\mathcal{U}(0.9, 1.0)$ | maximum performance |
| $\hat{\rho}$ | $\mathcal{U}(0.025, 0.1)$ | relative workload at the peak performance |
| $\bar{\rho}$ | $\mathcal{U}(0.25, 0.5)$ | relative workload at the inflection point of the decay phase |
| $k$ | $\mathcal{U}(0.05, 0.1)$ | steepness of performance decline |

Table 6: The range of parameters of human performance variation in Eq. (2) on FLickr10K dataset.

| Params | Range | Description |
|---|---|---|
| $w_0$ | $\mathcal{U}(0.65, 0.9)$ | initial performance |
| $w_{\text{base}}$ | $\mathcal{U}(0.3, 0.4)$ | minimum performance |
| $w_{\text{peak}}$ | $\mathcal{U}(0.8, 1.0)$ | maximum performance |
| $\hat{\rho}$ | $\mathcal{U}(0.025, 0.1)$ | relative workload at the peak performance |
| $\bar{\rho}$ | $\mathcal{U}(0.25, 0.5)$ | relative workload at the inflection point of the decay phase |
| $k$ | $\mathcal{U}(0.05, 0.1)$ | steepness of performance decline |

Table 7: The range of parameters of human performance variation in Eq. (2) on Micebone dataset.

| Params | Range | Description |
|---|---|---|
| $w_0$ | $\mathcal{U}(0.8, 0.9)$ | initial performance |
| $w_{\text{base}}$ | $\mathcal{U}(0.6, 0.7)$ | minimum performance |
| $w_{\text{peak}}$ | $\mathcal{U}(0.9, 1.0)$ | maximum performance |
| $\hat{\rho}$ | $\mathcal{U}(0.025, 0.1)$ | relative workload at the peak performance |
| $\bar{\rho}$ | $\mathcal{U}(0.25, 0.5)$ | relative workload at the inflection point of the decay phase |
| $k$ | $\mathcal{U}(0.05, 0.1)$ | steepness of performance decline |

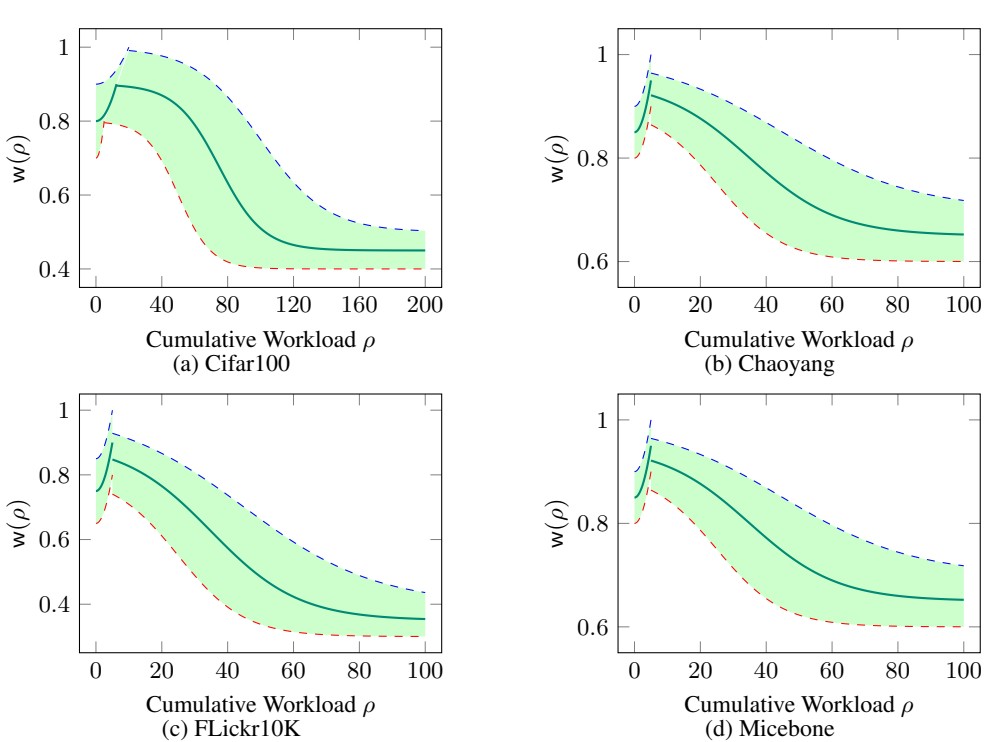

Figure 7: Human performance-Cumulative Workload curves on various datasets. The blue and red lines denote the upper and lower bound of human performance under cumulative workload accumulation.

