# OpenReview forum: "Fatigue-Aware Learning to Defer via Constrained Optimisation"
_ICLR.cc/2026/Conference — ICLR 2026 Conference Withdrawn Submission_

### Official Review · Reviewer_NXtg · 2025-10-29

**Soundness:** 3
**Presentation:** 3
**Contribution:** 3
**Rating:** 6
**Confidence:** 2

**Summary:**

The paper presents an important contribution to human-AI collaboration by formally integrating cognitive fatigue into learning-to-defer systems. The proposed FALCON framework is well-motivated and addresses a critical gap in existing literature: the assumption of static human performance. Its use of S5 for long-term state tracking and PPO-Lagrangian for constrained optimization demonstrates technical contributions.

**Strengths:**

- This work incorporates cognitive fatigue into the L2D paradigm. The deferral policy learns to avoid overloading humans during fatigue phases, improving system-level efficiency and human well-being.
- The human performance model draws from cognitive psychology and psychometrics, grounding the work in established science.
- The use of PPO-Lagrangian optimization with upper/lower cost constraints allows fine-grained control over human utilization, balancing automation and human engagement.

**Weaknesses:**

- Regarding the two-phase model, real human performance may exhibit non-smooth dynamics, recovery periods, or individual variability not captured by the sigmoid decay.
- The values of parameters (e.g., $w_0, w_{peak}, w_{base}$, etc.) need to be set carefully and estimated by each user.

**Questions:**

See the above weaknesses.

---

> ### Author Response · Authors · 2025-11-18
> **Non-smooth dynamics and individual variability**
>
> We acknowledge that real human performance can exhibit non-smooth dynamics, recovery periods, and individual variability. However, our goal is to provide a controlled and reproducible testbed for algorithmic development, focusing on the dominant and well-documented phenomenon of predictable fatigue accumulation. Cognitive psychology literature consistently shows that vigilance and accuracy decline over time during extended work sessions, following non-linear patterns (Sec. 2, lines 88–93; Waite et al., 2017; Reiner and Krupinski, 2012; Taylor-Phillips and Stinton, 2019; Estes, 2015; McCarley and Yamani, 2021). Our two-phase model (warm-up and sigmoid decay) reflects these established findings: initial skill acquisition followed by fatigue-induced decline. While recovery and individual variability are important, they do not invalidate the core phenomenon we model. We explicitly note in Sec. 7 that future work will incorporate instance-dependent fatigue and recovery dynamics, but the current design captures the primary effect ignored by all prior L2D methods.
>
> Moreover, FALCON is robust to variability. During training, we sample fatigue parameters from broad ranges (Tables 4–7, Fig. 6), creating thousands of distinct curves. This population-based strategy enables the policy to generalise across unseen patterns, as validated by strong zero-shot results (Fig. 4c, 4f, 4i). Unlike L2D-Pop and EA-L2D, which require per-expert context sets, FALCON adapts using only observed workload $\rho_t$, without retraining or fine-tuning.

---

> ### Author Response · Authors · 2025-11-18
> **Per-user parameter tuning**
>
> FALCON requires no per-user parameter tuning at deployment. During training, it enhances diversity by sampling fatigue parameters from broad ranges (Tables 4–7, Fig. 6), generating thousands of distinct performance curves to learn a robust policy. At deployment, the policy adapts solely based on observed workload $\rho_t$ (Algorithm 2, line 7), without any manual tuning. Unlike L2D-Pop, which relies on per-expert context configuration, FALCON works immediately with new experts. Zero-shot results (Figs. 4c, 4f, 4i) demonstrate that a single trained model effectively handles diverse, unseen experts.

---

### Official Review · Reviewer_44bJ · 2025-10-31

**Soundness:** 3
**Presentation:** 3
**Contribution:** 3
**Rating:** 6
**Confidence:** 2

**Summary:**

The paper proposes FALCON, a fatigue-aware learning-to-defer (L2D) framework for L2D that models human performance as workload-dependent using psychologically grounded fatigue curves and formulates L2D as a constrained MDP over sequences. It also introduces a  Fatigue-Aware L2D (FA-L2D) benchmark. Experiments on the benchmarks demonstrate the effectiveness of the propsoed FALCON.

**Strengths:**

1. The idea that incorporates workload-varying human performance and in a sequential decision-making task for L2D is well-motivated and interesting.

2. The paper introduces a Fatigue-Aware L2D (FA-L2D) benchmark, which faciliate the future exploration in the field.

3. Experiments demonstrates the effectiveness of the proposed method.

**Weaknesses:**

1. The method model human performance based on Eq. 2. However, it is not clear how to guarantee that the model of human performance is reasonable. With a human-in-the-loop or at a fit-to-real-data example may make the method more convincing.

2. In Eq. 2, modeling human performance relies on the introduced $\rho$, which is difficult to be explicit in real applications for real humans. It is not clear how does it make sense in real applications and how to make sure it is introduced precisely.

3. It is not clear whether the proposed method introduce extra training and inference costs compared to the baselines.

**Questions:**

I'm also curious how does the method be capable to achieve zero-shot generalization, any discussion or analyses on that?

---

> ### Author Response · Authors · 2025-11-18
> **Validating human performance model**
>
> We respectfully clarify that the fatigue model in Eq. (2) is not arbitrary; it is grounded in well-established findings from cognitive psychology on vigilance decrement and fatigue accumulation (Estes, 2015; Newell and Rosenbloom, 2013; McCarley and Yamani, 2021). Research in that area demonstrates that human accuracy predictably declines during extended work sessions, following non-linear patterns of warm-up and fatigue (Estes, 2015; Newell and Rosenbloom, 2013; McCarley and Yamani, 2021). For example, studies in radiology show that diagnostic errors increase significantly as fatigue accumulates (Waite et al., 2017; Reiner and Krupinski, 2012; Taylor-Phillips and Stinton, 2019). Our piecewise function captures these two documented phases: initial skill acquisition (warm-up) and sigmoid decay due to cognitive fatigue, both supported by empirical evidence in psychology and human factors literature (Sec. 2, lines 88–93).
>
> While real-world validation with human-in-the-loop is valuable, our simulation provides a controlled and reproducible testbed for algorithmic development, avoiding confounders such as health or environment. We explicitly acknowledge this limitation and commit to incorporating real human studies in future work (Sec. 7). Importantly, the current design reflects the fact that  cognitive fatigue is ignored by all prior L2D methods, making FALCON the first approach to address this critical realism gap.

---

> ### Author Response · Authors · 2025-11-18
> **Cumulative workload is difficult to be explicit in real applications for real humans**
>
> We appreciate the concern and clarify that the difficulty lies not in tracking workload $\rho_t$, which is straightforward in real applications (e.g., number of cases reviewed by a radiologist, items flagged for human review, or documents processed in a session), but in estimating the parameters $\hat{\rho}$, $\bar{\rho}$, $k$ that govern the shape of the fatigue curve. These parameters are not arbitrary; they correspond to interpretable psychological concepts:
>
> 1.  $\hat{\rho}$: relative workload at peak performance (end of warm-up phase)
> 2. $\bar{\rho}$: inflection point of fatigue-induced decline
> 3. $k$: steepness of performance decay
>
> These values can be estimated from historical data or short calibration studies in real deployments, similar to how ergonomic and cognitive workload models are parameterised in human factors research. Importantly, **FALCON is designed to be robust to misspecification**: during training, we sample these parameters from broad ranges (Tables 4–7, Fig. 6), creating thousands of distinct curves. This population-based strategy ensures the learned policy generalises to unseen or imperfectly specified fatigue patterns, as demonstrated by strong zero-shot results (Fig. 4c, 4f, 4i).

---

> ### Author Response · Authors · 2025-11-18
> **Training/inference costs**
>
> FALCON has comparable training and inference costs to baselines on CIFAR100 for training set about 1e7 iterations and test cases involving 50 episodes, as shown in Figure 5 and 6 (Appendix B.3).

---

> ### Author Response · Authors · 2025-11-18
> **Zero-shot generalization**
>
> In the training phase, FALCON increases training diversity by sampling fatigue parameters from broad ranges (Tables 4-7, Fig. 6), which creates thousands of distinct performance curves. In Sec.4 (lines 313-316), we discussed the zero-shot testing. *"Zero-shot Setting: This setting measures the ability to generalise to previously unseen human performance patterns. Methods use models trained on the main experiments with parameter ranges in Tables 4 to 7 and Fig. 6 without additional training or adaptation for the specific case being evaluated during testing."* Unlike L2D-Pop/EA-L2D, which requires few-shot expert context set, FALCON uses workload as a universal proxy for cognitive state. In zero-shot results (Fig. 4c, 4f, 4i), models trained on mixed distributions generalise to specific unseen curves with non-overlapping human performance coverage. The parameters of fatigue functions in training and testing are shown in Fig. 4 and Fig. 6a.

---

### Official Review · Reviewer_7e2s · 2025-11-02

**Soundness:** 3
**Presentation:** 3
**Contribution:** 2
**Rating:** 2
**Confidence:** 2

**Summary:**

This paper considers the human fatigue level in a Learning to Defer problem. It not only provides a model for the effect of human fatigue level on human decision accuracy, but also proposes a new CMDP method to solve it. Numerical results are also provided to illustrate the performance. Additionally, many data benchmarks and simulation environments are provided.

**Strengths:**

1. The paper proposed a new methodology for the fatigue-aware human-AI interaction problem.

2. The paper provides many data benchmarks for future research.

3. The numerical results look good.

**Weaknesses:**

1. From a methodology perspective, the paper might lack novelty. I might misunderstand the paper; however, it seems that the paper mainly transfers rho_t and x_t into some new features and then applies CMDP for a more stylized human reaction model. Could the authors elaborate more on the novelty of the methodology?

2. The human-reaction function form might be too simplified, which may oversimplify the complexity of the problem. Specifically, the function form is almost monotonic, given its short warm-up phase. If the human fatigue level can be recovered when no job is assigned for a while,  can this algorithm still solve the problem efficiently?

3. Even if the human-reaction form is correct, the numerical experiments might also be sufficient to prove the applicability of the proposed method, especially given that this paper does not have any theoretical guarantee. For example, what if the warm-up period is very long?

**Questions:**

1. What is the form of the backbone model?

2. Is the AI classifier trained together with the L2D learner?

3. What if the fatigue function is misspecified or different between training and test data? Can this algorithm recognize and correct it?

4. Given this setup, a simple idea is to learn the function between accuracy and the workload directly, and pass the approximation to a CMDP. Will this simple idea be much worse than the method proposed in this paper?

---

> ### Author Response · Authors · 2025-11-18
> **Novelty of FALCON**
>
> Thank you for this comment. We respectfully disagree with this characterization. FALCON's novelty consists of three fundamental contributions that go beyond simple feature engineering and the application of CMDP:
>
> 1. **FALCON is the first work to formulate L2D with time-varying human performance**: Standard L2D (Eq. 1) treats deferral as independent decisions with static human performance (e.g., humans are represented via the function $\mathsf{h}(x)$, which does not depend on temporal factors), while FALCON formulates L2D as CMDP where sequential decisions affect future states through $\rho_t$ (Eq. 4), requiring a fundamentally different optimisation strategy.
> As stated in Section 2.1: "*All existing L2D methods are built on the simplifying assumption that the performance of the human prediction $\mathsf{h}(x)$ is static over time, which is an assumption that ignores well-documented variations such as fatigue-induced degradation.*"
>
> 2. **CMDP Formulation with Budget Constraints**: We reformulate L2D as a Constrained Markov Decision Process (Sec. 3.3), enabling precise control over human–AI cooperation budgets while adapting to dynamic human states. Previous methods use static thresholds or unconstrained RL, which cannot guarantee coverage bounds.
>
> 3. **Psychologically Grounded Simulation Environment**:
> We develop a simulation environment based on cognitive psychology principles (Sec. 3.2), providing a realistic testbed for evaluating L2D methods under workload-variant human performance conditions.

---

> ### Author Response · Authors · 2025-11-18
> **Simplified Human Performance**
>
> We respectfully disagree that our human-reaction model is “simplified.” To the best of our knowledge, no prior L2D work has attempted to model dynamic human performance at all; existing methods make the unrealistic assumption that humans are static oracles. Our approach is therefore a first step toward realism, grounded in cognitive psychology literature that studies fatigue accumulation and vigilance decrement (McCarley and Yamani, 2021; Gyles et al., 2023; Newell and Rosenbloom, 2013; Estes, 2015). The piecewise fatigue curve in Eq. (2) reflects two well-documented phases:
> 1. Warm-up (skill acquisition): performance improves initially (Newell and Rosenbloom, 2013).
> 2. Fatigue (sigmoid decay): accuracy declines predictably with workload (Estes, 2015; McCarley and Yamani, 2021).
>
> This design is based on findings from psychological research and provides a controlled simulation environment for algorithmic development. While real-world performance involves additional confounders (expertise, health, environment), our model is the first in AI to capture the dominant factor of cognitive fatigue, which prior L2D models completely ignore.
>
> Regarding recovery, we note that within-session recovery is negligible compared to AI processing speed. The Berlin (2000) study shows a radiologist making critical errors after 162 cases in one day with minimal rest. Our episodic setup resets workload between sessions, implicitly modelling overnight recovery. Nevertheless, we acknowledge this limitation and plan to incorporate instance-dependent fatigue (Sec. 7) and recovery in future work.
>
> Finally, scenarios with long warm-up periods are covered in our experiments (Fig. 4a–c, “Sustained High Performance”), where human accuracy remains above 80\% throughout. FALCON still outperforms all baselines in both fine-tuning and zero-shot settings, demonstrating robustness even when fatigue is minimal. Moreover, our CMDP formulation builds on established RL theory (PPO-Lagrangian, Fujimoto et al., 2019), ensuring principled optimisation under constraints.

---

> ### Author Response · Authors · 2025-11-18
> **Backbone model**
>
> We mentioned the backbone model in Appendix B.1. For experiments performed on Cifar100 dataset, we employ PreAct-ResNet-18, while for other datasets, we train the AI model with a ResNet-18.
> After extracting the features from our backbone, we concatenate them to form workload embeddings, which are  processed by S5 layers for temporal modelling (Fig. 2b).

---

> ### Author Response · Authors · 2025-11-18
> **AI classifier training together with the L2D learner**
>
> The details of the training of the AI classifier and gating model are provided in Appendices B.1 and B.2, respectively. Following prior works, we train the AI model and the L2D gating model separately. Only the gating policy is trained via PPO-Lagrangian.

---

> ### Author Response · Authors · 2025-11-18
> **Misspecified fatigue function**
>
> FALCON is inherently robust to fatigue-function  misspecification. During training, each episode samples fatigue parameters from broad ranges (Tables 4–7, Fig. 6), generating thousands of distinct performance curves.  This diversity acts as a population-based training strategy, ensuring the learned policy generalises to unseen or misspecified fatigue patterns at test time.
> We explicitly evaluate this in zero-shot settings (Sec. 4, Figs. 4c, 4f, 4i). FALCON generalises well to unseen fatigue patterns without retraining, outperforming all baselines. This robustness stems from modelling fatigue as part of the state and learning adaptive policies rather than hard-coded thresholds.

---

> ### Author Response · Authors · 2025-11-18
> **Simple baseline alternative**
>
> The proposed alternative of directly learning an accuracy–workload function assumes stationarity and perfect estimation, which is unrealistic in dynamic environments where human performance is unknown, instance-dependent, and varies across experts.
> Such an approach would require retraining for every new fatigue pattern, whereas FALCON learns adaptive policies that generalise across diverse curves by sampling parameters from broad ranges during training (Tables 4–7, Fig. 6), enabling robust zero-shot performance (Fig. 4c, 4f, 4i).
> Our ablation studies (Sec. 5) show static approximations degrade significantly under rapid fatigue, while FALCON maintains high accuracy.
> Moreover, counting-based strategies like EA-L2D fail to scale with large class sets (Sec. 5, lines 368–371), underscoring the limitations of simple mappings compared to our CMDP-based approach with PPO-Lagrangian optimisation.

---

### Author Response · Authors · 2025-11-18
**Changes made in the revision**

We thank all reviewers for their insightful feedback. We have carefully considered the comments from each reviewer and have incorporated these suggestions into a revised version. All modifications are highlighted in blue in our paper. Specifically, we have:

- add the training time of FALCON and competing methods on Cifar100 (Figure 5 in Appendix)

We have also prepared detailed responses to each reviewer's individual concerns, which can be found directly below their respective reviews.

---

### Comment · Area_Chair_Difi · 2025-11-25

Dear Reviewers

Thank you for your time and help for reviews.
Authors have uploaded their rebuttals. If you have not done yet, please review the authors' rebuttal for the paper under your evaluation and engage in discussion with authors.

Thank you again.
Best,

Area Chair

---

### Author Response · Authors · 2025-12-01
**Summary of Reviewers' Concerns and Responses**

We have comprehensively addressed all reviewer concerns by clarifying our contributions, justifying our psychologically grounded model with extensive literature support, and demonstrating robustness through population-based training and zero-shot evaluation.

Below is a summary of our responses:

**Novelty and Contribution (7e2s)**

We clarify FALCON's novelty consists of three fundamental contributions:

- **FALCON is the first work to formulate L2D with workload-variant human performance**: Standard L2D (Eq. 1) treats deferral as independent decisions with static human performance (e.g., humans are represented via the function $\mathsf{h}(x)$, which does not depend on workload accumulation), while FALCON formulates L2D as CMDP where sequential decisions affect future states through $\rho_t$ (Eq. 4), requiring a fundamentally different optimisation strategy.

- **CMDP Formulation with Budget Constraints**: We reformulate L2D as a Constrained Markov Decision Process (Sec. 3.3), enabling precise control over human–AI cooperation budgets while adapting to dynamic human states. Previous methods use static thresholds or unconstrained RL, which cannot guarantee coverage bounds.

- **Psychologically Grounded Simulation Environment**:
We develop a simulation environment based on cognitive psychology principles (Sec. 3.2), providing a realistic testbed for evaluating L2D methods under workload-variant human performance conditions.

**Validity of Human Performance Model (7e2s, 44bJ, NXtg)**

Our approach is grounded in cognitive psychology literature that studies fatigue accumulation and vigilance decrement (McCarley and Yamani, 2021; Gyles et al., 2023; Newell and Rosenbloom, 2013; Estes, 2015). The piecewise fatigue curve in Eq. (2) reflects two well-documented phases:
1. Warm-up (skill acquisition): performance improves initially (Newell and Rosenbloom, 2013).
2. Fatigue (sigmoid decay): accuracy declines predictably with workload (Estes, 2015; McCarley and Yamani, 2021).

This design is based on findings from psychological research and provides a controlled simulation environment for algorithmic development. While real-world performance involves additional confounders (expertise, health, environment), our model is the first in AI to capture the dominant factor of cognitive fatigue, which prior L2D models completely ignore.

Regarding recovery, we note that within-session recovery is negligible compared to AI processing speed. Our episodic setup resets workload between sessions, implicitly modelling overnight recovery. Nevertheless, we acknowledge this limitation and plan to incorporate instance-dependent fatigue (Sec. 7) and recovery in future work.

FALCON is inherently robust to fatigue-function diversity and requires no per-user parameter tuning at deployment. During training, each episode samples fatigue parameters from broad ranges (Tables 4–7, Fig. 6), generating thousands of distinct performance curves. This diversity acts as a population-based training strategy, ensuring the learned policy generalises to unseen or misspecified fatigue patterns at test time. We explicitly evaluate this in zero-shot settings (Sec. 4, Figs. 4c, 4f, 4i). FALCON generalises well to unseen fatigue patterns without retraining, outperforming all baselines. This robustness stems from modelling fatigue as part of the state and learning adaptive policies rather than hard-coded thresholds.

Finally, scenarios with long warm-up periods are covered in our experiments (Fig. 4a–c, “Sustained High Performance”), where human accuracy remains above 80\% throughout. FALCON still outperforms all baselines in both fine-tuning and zero-shot settings, demonstrating robustness even when fatigue is minimal.

**Practical Implementation (44bJ, NXtg)**

The difficulty lies not in tracking workload $\rho_t$, which is straightforward in real applications (e.g., number of cases reviewed by a radiologist, items flagged for human review, or documents processed in a session), but in estimating the parameters $\hat{\rho}$, $\bar{\rho}$, $k$ that govern the shape of the fatigue curve. These parameters are not arbitrary; they correspond to interpretable psychological concepts:
- $\hat{\rho}$: relative workload at peak performance (end of warm-up phase)
- $\bar{\rho}$: inflection point of fatigue-induced decline
- $k$: steepness of performance decay


**Baselines, Cost, and Architecture (44bJ)**

*Computational Cost*: FALCON has comparable training and inference costs to baselines, as shown in Figure 5 and 6 (Appendix B.3).

*Architectures*: We clarified that we use standard backbones (PreAct-ResNet-18 for CIFAR100, ResNet-18 for others) and that the AI classifier and gating models are trained separately, consistent with prior L2D literature.

---

### Note · Authors · 2026-01-27

I have read and agree with the venue's withdrawal policy on behalf of myself and my co-authors.

---

### Meta-Review · Area_Chair_rTJ8 · 2025-12-16

**Summary:**

This submission receives 2, 6, 6 scores. All three reviewers gave low-confidence level (2). There are the following major concerns:

* Lack of Novelty (Reviewer 7e2s): A key concern was that the methodology might lack novelty, potentially boiling down to simple feature transfer ($\rho_t$ and $x_t$) combined with the application of Constrained Markov Decision Process (CMDP).

* Simplified Human Performance Model (Reviewer 7e2s, NXtg): The two-phase human-reaction function (warm-up and sigmoid fatigue) was seen as potentially oversimplifying the complexity of real human performance. Concerns included the model being almost monotonic given the short warm-up, and the lack of modeling for non-smooth dynamics, recovery periods, or individual variability.

* Difficulty in Parameter Estimation (Reviewer 44bJ, NXtg): Reviewers questioned the practicality of estimating the fatigue curve parameters ($\hat{\rho}, \bar{\rho}, k$) in real-world applications for individual users, noting that the cumulative workload ($\rho_t$) is difficult to explicitly define for real humans.

* Model Validation (Reviewer 44bJ): There was a concern that the human performance model's reasonableness was not sufficiently guaranteed, suggesting that a human-in-the-loop experiment or a fit-to-real-data example would make the method more convincing.

* Insufficiency of Experiments (Reviewer 7e2s): Without a theoretical guarantee, the numerical experiments might be insufficient to prove applicability, particularly for scenarios like a very long warm-up period.

AC has read the paper and gone through the reviews and discussion. The paper addresses a critical and novel problem in Learning to Defer by incorporating human fatigue. The CMDP formulation and the resulting zero-shot generalization capabilities are technically sound and innovative. However, the work's dependence on a purely simulated, psychologically grounded but empirically unvalidated human performance model is a major limitation. The lack of an ablation against a simpler modeling baseline and the absence of any human data (even a small-scale fit) severely restricts the confidence in its practical applicability. Thus, a resubmission is suggested to include further the critical empirical validation against real human performance data.

**Reviewer Concerns:**

Addressed by the rebuttal:
* Concerns on Novelty and Contribution: FALCON's novelty is in three areas: 1) It is the first L2D work to formulate the problem with workload-variant human performance, requiring a sequential CMDP approach where deferral affects future states through $\rho_t$. 2) It uses a CMDP formulation with budget constraints (PPO-Lagrangian) to control human utilization precisely. 3) It provides a psychologically grounded simulation environment.

* Concerns on Simplified Human Model: The model is a first step toward realism and is grounded in established cognitive psychology literature (warm-up and sigmoid fatigue). They note that recovery is negligible within a session, and the episodic setup models overnight recovery. Future work will incorporate instance-dependent fatigue and recovery.

* Concerns on Parameter Estimation: The difficulty is in estimating the shape parameters, not tracking the workload $\rho_t$ (e.g., number of cases reviewed). Parameters can be estimated from historical data or short calibration studies. Critically, FALCON requires no per-user tuning at deployment due to its robust zero-shot generalization.

Outstanding concerns:
* Necessity of the Complex Model (7e2s): Reviewer 7e2s proposed a simpler baseline: learning an accuracy-workload function and passing the approximation to a CMDP. The authors defended their approach by arguing this simple baseline requires perfect estimation and retraining. However, they did not provide a direct empirical comparison to show their method is substantially better than a well-tuned simpler alternative, which is often crucial when introducing complexity.

* Model Validation with Real Data (44bJ): While the model is psychologically grounded, Reviewer 44bJ requested a human-in-the-loop experiment or a fit-to-real-data example to guarantee the reasonableness of the model in a real-world setting. The authors acknowledged this as a limitation for future work, but the lack of any real-world data validation weakens the claim of practical applicability.

**Reviewer Scores:**

4, 6, 6

---

### Decision · Program_Chairs · 2026-01-26

Reject